# Review: Interactions of Active Colloids with Passive Tracers

**Linlin Wang and Juliane Simmchen \***

Institute of Physical Chemistry, TU Dresden, Zellescher Weg 19, 01062 Dresden, Germany
* Correspondence: Juliane.Simmchen@tu-dresden.de

**Abstract:** Collective phenomena existing universally in both biological systems and artificial active matter are increasingly attracting interest. The interactions can be grouped into active-active and active-passive ones, where the reports on the purely active system are still clearly dominating. Despite the growing interest, summarizing works for active-passive interactions in artificial active matter are still missing. For that reason, we start this review with a general introduction, followed by a short spotlight on theoretical works and then an extensive overview of experimental realizations. We classify the cases according to the active colloids' mechanisms of motion and discuss the principles of the interactions. A few key applications of the active-passive interaction of current interest are also highlighted (such as cargo transport, flow field mapping, assembly of structures). We expect that this review will help the fundamental understanding and inspire further studies on active matter.

**Keywords:** active colloids; active-passive interaction; collective phenomena

## 1. Introduction

Synthetic active matter has become a core interest in physics due to its ability to emulate living systems. One of the more interesting and challenging aspects is synergies like communication and cooperation, which fundamentally rely on interactions between active agents [1]. There is a vast amount of theoretical papers on interactions and also a large number of experimental findings, with a strong focus on interactions between active particles (including phase transitions, etc.), which have been treated extensively [2,3] and will not be discussed here in depth. Excellent reviews on the general topic have been published over the last few years [2,4–10]. They associate the origin of the interactions either with chemical, hydrodynamic, medium-generated or electromagnetic dipole interactions, or using the authors' word, "communication" [3]. Keeping these categories in mind, we refer the reader to these works and the references therein. Another way to see these assemblies is to consider them as building blocks or "modules" to construct more complex and functional swimmers. Here, not only possible assemblies between actives are discussed, but also formations including passive objects [11].

However, looking at biologically-active particles like sperm cells, algae, bacteria, or motor proteins, they encounter other active particles of their same kind regularly, but much more frequently, they encounter non-active obstacles. Examples could be polymers in mucus or extracellular matter, vesicles within cells, or dead bacteria in biofilms. These mixed species or active passive interactions affect the swimmers' behaviors widely and have therefore also become a core interest of the active matter community. Lately, we can observe an increase of theoretical predictions, as well as experimental observations. The goal of this manuscript is to give the reader an introduction to the topic and the theory and subsequently a systematic, extensive overview about the experimentally-observed interactions between active and passive particles, separated into groups of individual micromotors, divided by their propulsion mechanisms. The manuscript concludes with a brief discussion and a display of possible applications of these interaction phenomena.



## 2. Spotlight on Theoretical Considerations

In order to keep this review focused, we will just give a brief overview of the theoretical approaches followed to explore interactions; for an in-depth discussion, a separate work is necessary. To induce motility on the microscale in biology, different strategies are employed, ranging from body deformation to surface gliding. Living systems, but also artificial micromachines, constantly transform some kind of energy into kinetic energy and are therefore constantly out of equilibrium. This requires analytical and numerical tools from nonequilibrium statistical physics for their description.

One often followed approach is "dry active matter". Here, one considers a fluid made out of active particles without any suspending fluid in-between. For single active particles, the dynamics are described using the Langevin equation, and the motility can be derived considering also the statistical distribution of run-and-tumble particles [12]. However, considering individual swimmers, this approach is limited to single particles and does not capture induced flows by additional particles or the presence of walls.

One frequently-used model in the field is the Active Brownian Particle (ABP), which similar to several other models, focuses solely on the active particles and completely disregards hydrodynamic interactions with the surroundings. At large spatial and temporal scales, a collection of such ABPs leads to an "active fluid" described in terms of fields, i.e., the particles themselves can behave as a fluid and generate interactions. This was applied to observe motility-induced instabilities in active-passive mixtures, and despite not giving insights into the microscopic mechanism, the simulations showed behaviors reminiscent of some experimental observations [13].

An extension of the ABP was recently presented by the combination with long-range interactions of chemical (phoretic) origin. Which interactions were expected to be the dominating ones among active particles were discussed, identifying three types of interactions: steric, hydrodynamic, and phoretic ones [14,15]. Their minimal description of an active attractive alignment model reduces the phoretic interactions to a simple pair interaction, making the concept simple enough to be used in ABP models. Whether this assumption is valid for certain cases has raised some ongoing discussions [16].

Furthermore, assuming that the hydrodynamics are of a much shorter range than phoretic interactions [17], modeling the activity of the active particles as a constant velocity and creating a chemical sink field around the particles was shown [18]. Separate overdamped Langevin equations for active and passive tracers yield the positon of the passives, and besides emerging dynamics (gas-like, oscillating, and collapsing), the model captures well the decreasing speed of growing clusters observed experimentally [18].

The second concept is "wet active matter". Ground-breaking work on the detailed mechanisms of the motion of individual particles, so-called "squirmers" was published by Lighthill in 1952 [19]. "Squirmers" propel themselves by a prescribed axisymmetric surface velocity field. This model of mechanical interactions with the surroundings was refined and adapted to match *Paramecium* [20]. Increasingly, artificial systems are also capable of moving in this bio-inspired way [21]. Here, the active particles are suspended in a fluid, and the hydrodynamics occurring are captured by the Navier–Stokes equations, which simplifies to a linear, time-independent equation due to the low Reynolds number.

$$\nabla p + \eta \nabla^2 u = 0$$

$$\nabla \cdot u = 0.$$

The hydrodynamic interactions [22,23] refer to the flow fields created by the particle motion that influence neighbors and the surroundings. This is very useful for most biological swimmers, where the absence of chemical solutes limits the origin of motion to hydrodynamic influences, which are approximated well by a force dipole. De Graaf et al. used a force-counterforce-based lattice Boltzmann algorithm to capture the low Reynolds number physics, reproducing the results very well in the far field [24]. In the vicinity of the bacterial swimmers, the authors found that the interactions were more likely to be disturbed, so no precise predictions for the close fields were to

be expected from this approach. From a different point of view, namely the influence of passive particles on the swimming speed of bacteria considering non-attractive interactions among these agents, Zoettl and Yeomans reported the influence of colloids on the mean position of the bacteria [25]. This approach considers both the influence of the squirmer on the surrounding fluid, as well as the reverse effect back on the active swimmer. Interestingly, it also confirms the absence of "stickiness" for hydrodynamically-propelled biological microswimmers.

For many artificial swimmers, the assumed mechanism relies on different physicochemical fields, produced by a variety of reactions on one spot on the particle surface. The solutes that are generated by the reactions are distributed dynamically around the active colloids, interact with the particle, and drive an interfacial fluid flow, leading to the particle propulsion. In the commonly-applied models, this behavior is displayed using the Laplace equation to describe the steady state of the solute concentration:

$$\nabla^2 c(r) = 0,$$

Near the surface of the particle, there is a force acting on the fluid, which is caused by the solutes interacting with the surface. This force drives flow around the particle, but is typically replaced by a slip flow boundary condition. In general, the flow field of motion can be described by a combination of singularities. Often used is the combination of a force dipole (flow fields decay with $1/r^2$), a source dipole, and a force quadrupole (decaying with $1/r^3$), which, when superposed, describe the flow field with $1/r^3$ accuracy. However, this general rule has many exceptions due to the variety of mechanisms. Considering that chemical gradients are crucial for self-diffusiophoresis, it becomes evident that besides influencing the dynamics of swimming, they are felt by walls and passive particles. This causes a response, similar to the collective motions of many motors, i.e., through self-generated concentration gradients and the gradients of other motors in the vicinity. Furthermore, for chemical swimmers, purely a hydrodynamic squirmer-based model has been discussed [26]. The authors suggested to use interactions with the surroundings to differentiate between different flow field patterns around the squirmers. Based on that, Popescu et al. showed that in the case of a collision between two identical model active particles, it is not valid to replace the active particle by its effective squirmer since this would mean attributing the interaction completely to hydrodynamic effects. The authors even highlighted the risk of misinterpreting observed effects to use them for the classification of microswimmers as squirmers [27]. This has been also taken into consideration by others who contrasted the effect of chemical interactions on factors such as the radial distribution function and the velocity [28,29].

Another difficult problem is the consideration for dense suspensions, since here, a mere superposition of individual swimmers is not suitable, and complex multibody effects have to be considered. However, the question of how much simplification of the system is valid and helps economizing computational time and when important aspects start to be neglected is an ongoing dispute. This extensive interest in interactions often considers only active-active interactions, sometimes in confinement [30], but can be largely extrapolated to the influence on passive particles.

From the theoretical side, interesting insights into and contributions to the understanding of the microscopic mechanism are to be expected and are needed to identify key factors to enable experimentalists to understand, control, and tune individual aspects of these effects.

## 3. Experimental Studies

### 3.1. Body Deformation

Even though biological systems are not the core of the study, we can get insights from looking at them. Wu and Libchaber compared the incessant motion of 4.5- and 10-μm particles in a fluid to their behavior in a bacterial bath in a 2D confined geometry, achieved by a thin soap film, which strongly reduced the hydrodynamic dissipation compared to bulk solutions [31]. The particle behavior was found to be more strongly dependent on the *Escherichia coli* concentration rather than the bead size.

The comparison between *E. coli* and active electrophoretic rods gave similar findings despite the clearly different motion mechanisms. Both moved close to the surface, and individual swimmers displayed different trajectories and without causing significant attractive interaction or "stickiness" on the passive colloids, but both resulted in an enhanced diffusion for these [32]. The absence of "stickiness" was also confirmed by Makarchuk et al., who additionally found an influence on the bacteria's trajectories [33]. Tracer trajectories have been obtained, but since the here simulated swimmers were inspired by biological swimmers (*E. coli* for pushers and *Chlamydomonas* as pullers), no clustering of tracers in the surrounding of the swimmers has been observed. These experimental findings have been theoretically reproduced by De Graaf et al., who managed to simulate the trajectories of tracers that were advected in the flow field of non-tumbling bacterial microswimmers very well [24]. Similar results have been reported considering different theoretical [25] and experimental studies of bacteria in polymers or colloidal baths and have also been confirmed for other types of biological swimmers [34,35]. Concluding, none of the experimental tracer studies with biological microswimmers found significant stickiness.

### 3.2. Phoresis

One of the most explored propulsion sources is self-phoresis, which relies on the formation of a local gradient [36], for example of chemical or electrical potential. These can induce a locally-confined interaction of a species in solution with the surface of the colloid, happening in a thin interfacial layer. The nature of the gradient can be either the concentration of solute, electric charges, or thermal energy (temperature). The respective species would then cause local interactions, resulting in an assumed "phoretic slip velocity", which is defined as a thin layer interaction. Differently from this, we also have to consider the phoretic interaction, affecting other particles in the vicinity, being a phoretic response to the external gradient produced by an active particle.

### 3.2.1. Diffusiophoresis

We start our considerations with the simple case of a solute concentration gradient, which is oxygen in most cases. Starting from 2010, Sen's group published observations on AgCl colloids, which also interacted with passive tracers [37], but since the diffusing species in this system is most likely chloride ions, the categorizing of this system is not quite clear. Additionally, no individual interactions were considered since only collective motion was studied. More recently, further works on similar systems were published [38,39], and a Janus particle-based study found repulsive interactions caused by the product gradient when exposed to passive tracer particles; also here, the classification of the motion as diffusiophoresis was questionable, due to the ionic nature of the solutes. In the most intensely-studied system of Pt-capped Janus particles in dilute peroxide, earlier observations included interactions [40,41], but rather than investigating mechanisms, these were presented as the means for cargo transportation. Even in more recent work, interactions have been regularly observed and discussed, but few papers have been published to this moment. Several people reported the interactions with passive particles as ongoing work [42,43] and shared the observation that passive tracers are attracted to the inert side of the particle, leaving an exclusion zone surrounding the active hemisphere.

Interestingly, changing the material of the catalytic cap to copper or silver, materials that engage in the degradation of peroxide, not only the direction of motion was reversed. The interactions with passive colloids can be switched from strongly interactive to repulsive, where we observe the repulsive interactions to be more long range [43]. We assume this to be an indicator of the importance of the nature of the chemical gradient, but the proof is still ongoing work.

Campbell et al. were able to compare predicted flow fields around active particles to platinum Janus particles stuck to the substrate [44]. No detailed explanation was given on how the authors suppressed the raft formation observed by us and others, but most likely, they used a very low tracer concentration, imaging over long time periods. Besides the necessary size difference (which needs to be maintained in order to minimize the disturbance of the flow field due to the presence of the

tracer), we found it difficult to assess information of the bare swimmer flow field due to the intense raft-forming interactions.

Apart from the interactions between active and passive particles, which are based on the gradient produced by the active particle itself, the interactions can be enhanced further when using surface-modified particles. For instance, Gao and coworkers reported hydrophobic octadecyltrichlorosilane (OTS)-modified Janus micromotors [45]. The authors claimed that the defined active motor assemblies were formed due to hydrophobic interactions between OTS-modified hemispheres of motors, and there was no assembly behaviors between motors without hydrophobic coating. Moreover, assemblies of motor–non-motor (active-passive) could also be formed due to the hydrophobic interaction between OTS-modified motor hemispheres and neighboring passive particles. As shown in Figure 1A, various structures (doublet, triplet, and quadruplet motor–non-motor assemblies) were assembled with different amounts of active motors and passive particles, which minimized the potential energy. Well-defined assemblies with multiple passive particles could also be obtained through hydrophobic interactions (Figure 1B). However, the authors made no comparison to the interactions between active and passive particles without surface functionalization, so a clear distinction cannot be drawn.

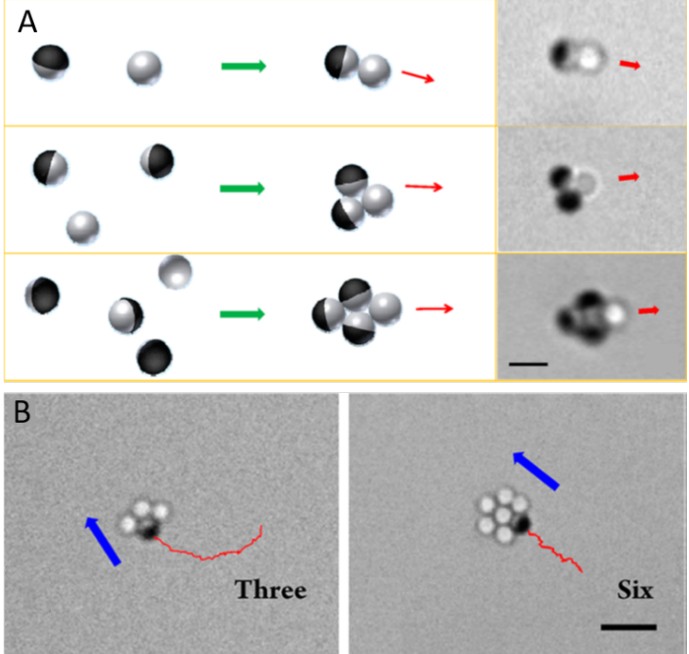

**Figure 1.** (**A**) Assemblies of OTS-modified Janus motor–non-motor particles, schematic (**left**) and microscopic images (**right**) (doublet, triplet, and quadruplet). Scale bar, 2 μm. (**B**) Organized multiple passive particles (three and six) assemblies and transport with one active motor (microscopic images). Scale bar, 5 μm. Reproduced with permission from [45]. Copyright 2013 by the American Chemical Society.

### 3.2.2. Light-Induced Phoresis

This part needs to be located within the diffusiophoretic micromotors, but deserves a special treatment since light-induced reactions offer several special benefits compared to fuel-catalyzed reactions: light is easily switchable, and a wireless energy transmission is possible. One of the earlier light-driven systems was Palacci's hematite composites [46], available in different shapes and compositions. Early on, the interactions with passive particles were also being considered, as in the example of the "peanut-shaped" hematite colloidal dockers, which are typically ~1.5 μm long and ~0.6 μm wide, propelling preferentially perpendicular to their long side (Figure 2A). By changing the pH value and the tetramethylammonium hydroxide content of the solutions, the author could switch

between attractive and repulsive interactions and therefore efficiently load, transport, and release colloidal cargoes [46]. Similar effects have been observed by the group of Pietro Tierno, also using active hematite colloids and passive tracers [47]. The authors denominated the interactions as "phoretic attraction" and considered the hematite colloids as sources. They simplified the chemical field experienced by the passives to a dipolar chemical field, inducing a chemical mobility in both actives and passives. The authors found a stronger attraction between active and passive particles, which can be ascribed to magnetic interactions. Additionally, two actives can form dimers and subsequently larger structures, resulting in the formation of gel-like structures (Figure 2B). Another well-known photoactive semiconductor is titanium dioxide, which has recently been the base of a large number of micromotors, but the discussion will be limited to these publications considering active passive interactions. In 2016, Singh et al. reported a system of silica particles, half covered with a thin titania layer that propel efficiently in dilute peroxide solutions (Figure 2C). Similar to the work of Palacci, here, the pH also strongly influenced the observations: working at a constant, close to neutral pH, attractive, light intensity-dependent interactions were observed. However, upon lowering the pH, passive particle attraction was not observed due to a reversal of the phoretic mobility on $SiO_2$, causing the repulsion of the passive $SiO_2$ particles [48]. Probably the most pronounced effects were observed in a report by Wang et al. [49]. Different fuel and light conditions enabled the authors to tune the interactions of copper-capped titania Janus particles from repulsive in dilute peroxide with visible light, to slightly attractive in pure water with UV light, as shown in Figure 2D. Very strong attractive interactions were observed when the colloids showed very fast motion, so that up to six layers of passive particles could be transported along with the active colloid. Further investigations proved the robustness of the effect using a broad range of materials and shapes of passive particles [50].

### 3.2.3. Electrophoresis

To differentiate between the influence of the flow and the electric fields created by micropumps, Esplandiu et al. used a variety of tracers that have different interactions with the fields. Even though here, no motile particles were used, the effect of the phenomena on different passive particles was visualized nicely. Charged passive tracers respond more strongly to the electric field created by the micropump, while neutral colloids interact dominantly with the hydrodynamic flow [51]. For bimetallic nanorods, due to the bipolar decomposition of hydrogen peroxide, the gradient of protons results in an electric field, thus leading to motion (electrophoresis) [52,53]. Despite the fact that electrophoretic motion does not seem to cause significant phoretic interaction [32], by modifying the bimetallic nanorods with certain materials, target cargoes are attracted. For instance, as shown in Figure 3A, positively-charged polystyrene particles (PS) are attracted by negatively-charged PPy via electrostatic interaction. Streptavidin-functionalized PS cargo could be attracted via specific biotin–streptavidin interaction [52]. Another way to understand the self-generated electric field is using more tracer particles. As shown in Figure 3B, when mixing the rod-shaped motors with charged tracer particles, negatively-charged particles are attracted by the Pt end, while positively-charged species migrate to the Au end. The position and charge of passive tracers can show that the local electric field is caused by Pt to Au, resulting from the proton generation on the Pt surface and their consumption at the Au end. Different from pure electrostatic interaction, this behavior is also based on electrophoretic force, which is caused by the self-generated field [53].

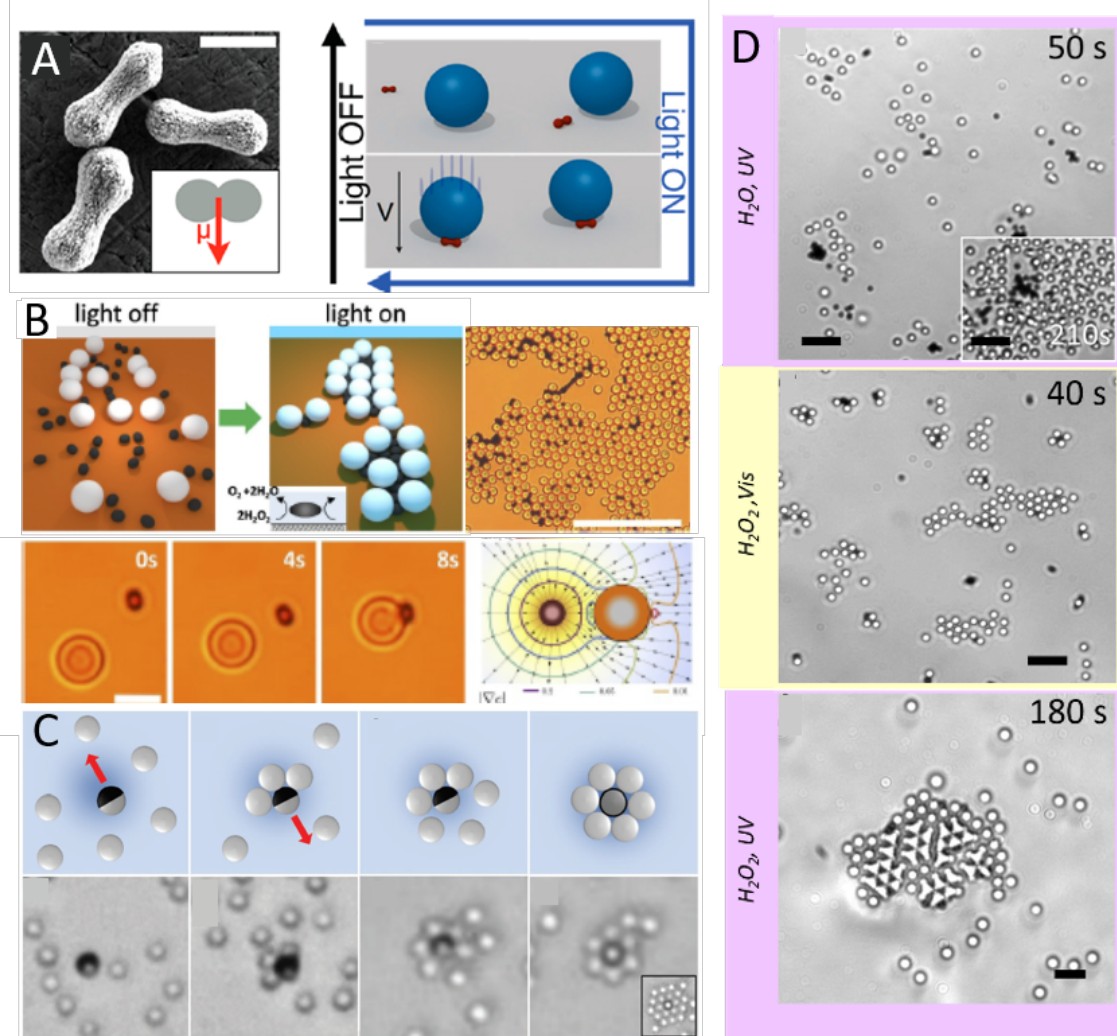

**Figure 2.** (**A**) SEM picture of the hematite peanut particles. Scale bar, 1 μm. The inset shows the direction of the permanent magnetic moment μ of the particle, perpendicular to the long axis. Schematic of docking. The hematite particle docks with the passive sphere and transports it with the steering of a magnetic field when the light is on. Reproduced with permission from [46]. Copyright 2013 by the American Chemical Society. (**B**) First row: Schematic and microscopic images showing the assembly of colloidal gel due to phoretic attraction between passive (white) and active particles (black) in the application of blue light. Scale bar, 50 μm. Second row: Microscope images showing the light-induced attraction between an active-passive pair of particles (left) and the color-coded logarithm of the chemical concentration around the pair (right). Reproduced with permission from [47]. Copyright 2018 by National Academy of Sciences of the United States of America (NAS). (**C**) Schematic (first row) and microscopic images of the assembly process from active Janus ($SiO_2$-$TiO_2$) and passive particles through the attractive interactions induced by light. The inset to last image is the cluster at maximal UV intensity. Reproduced with permission from [48]. Copyright 2017 by WILEY-VCH. (**D**) Different interaction behaviors of active Janus particles (Cu@$TiO_2$) with passive particles in different light and fuel conditions ($H_2O$ + UV, $H_2O_2$ + Vis, $H_2O_2$ + UV). They show repulsive interaction in dilute peroxide with visible light, and that can be tuned to slightly attractive in pure water with UV light. Especially, they show strong attraction in peroxide with UV light. Reproduced with permission from [49]. Copyright 2018 by the Royal Society of Chemistry.

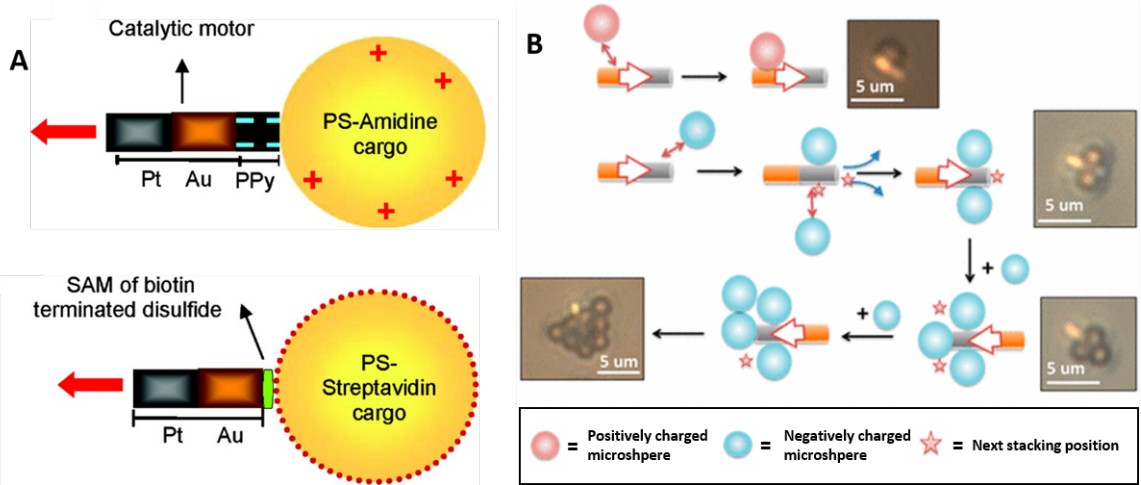

**Figure 3.** (**A**) Attraction via electrostatic interaction between the negative PPy end of a Pt-Au-PPy motor and a positively-charged PS particle and biotin–streptavidin interaction between the modified nanorod and cargo (SAM of biotin terminated disulfide, self-assembled monolayer of biotin-terminated disulfide). Reproduced with permission from [52]. Copyright 2008 by the American Chemical Society. (**B**) Scheme of passive cargo attraction via electrophoretic force between the Pt-Au motor and positively- and negatively-charged particles. Reproduced with permission from [53]. Copyright 2013 by National Academy of Sciences of the United States of America (NAS).

### 3.3. Induced Charge and Self-Dielectrophoresis

Different from micromotors that are based on self-electrophoresis, under an external alternating current (AC) electric field, metallodielectric Janus particles can realize self-propulsion with two mechanisms: induced charge electrophoresis (ICEP) [54–56] and self-dielectrophoresis (sDEP) [55–57] altered by the frequency of the electric field. At low frequencies, an asymmetric induced-charge electroosmotic flow (induced by different polarizabilities of two hemispheres) leads to motion in the direction of the dielectric side (ICEP) [54–56]. At high frequencies, beyond the relaxation time of the induced electric double layer, Janus particles are driven by a dielectrophoretic force arising from localized electric field gradients generated between Janus particle and the substrate (Figure 4A), moving with their metallic hemisphere forward (sDEP) [55–57]. Simultaneously, the frequency also dominates the mode of interaction with passive particles. Normally, the interactions between those Janus metallodielectric particles and dielectric particles are due to dielectrophoretic interactions between them, which are induced by the different polarizability compared to the medium [55–59]. Passive particles are attracted to regions with high field strength when they are more polarizable than the surroundings, namely positive dielectrophoresis (pDEP), otherwise they are repelled (negative dielectrophoresis (nDEP)). At low frequencies, passive particles are behind the motors and around the middle line due to the combination of pDEP and induced-charge electroosmotic flow (Figure 4(Bb)). In contrast, at high frequencies, passive particles are attracted to the front of the motors (due to pDEP) since the swimmers changed their motion direction (Figure 4(Bc)) [56]. Noticeably, at high frequencies, pDEP transits to nDEP by aligning the frequency, which is higher than the crossover frequency (COF); thus, passive particles will switch from being attracted to being repelled. Since the COF is dependent on particle properties (such as size), differently-sized passive particles could be selectively attracted or repelled by altering frequencies (Figure 4B; the highlighted green area is selective). Corresponding to the highlighted green area, Figure 4(Bd) is the selection stage, i.e., the bigger particles are repelled, while the smaller are attracted [55,56]. With a Janus system composed of different materials (Pt/SiO$_2$), Demirörs et al. have also studied the interaction between active Janus and passive tracers under high frequency fields (Figure 4C) [57]. They demonstrated that the active-passive shuttle's motion velocity scales quadratically with respect to the applied electric field strength **E** .

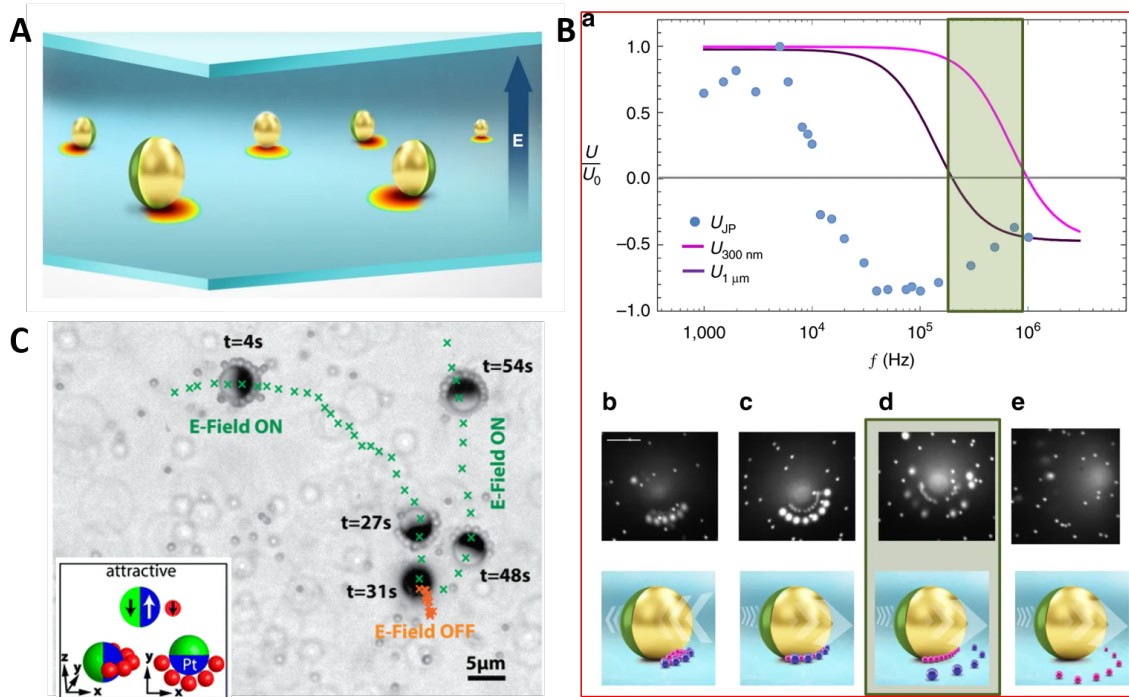

**Figure 4.** (**A**) A local electric gradient is induced around the polarizable surface of Janus Au/PS particles. (**B**) Selective cargo assembly and transport. (**a**) Precise control of the frequency of the applied field enables selective transport with different sizes of PS passive particles (magenta, 300 nm; purple, 1 μm). In the green highlighted region, bigger particles are repelled, while smaller particles are attracted. Four frequency domains results in the selectivity. Microscope images and schematics below. (**b**) At low frequencies, the Janus particle translates forward under induced charge electrophoresis (ICEP). Both the small and big particles undergo positive dielectrophoresis (pDEP); there is non-selective trapping. (**c**) With increasing frequency, the Janus particle reverses direction, but both small and big particles still undergo pDEP. (**d**) Selective trapping; smaller particles still undergo pDEP (attracted), while bigger particles undergo negative dielectrophoresis (nDEP) (repelled). (**e**) At higher frequencies, small target particles undergo nDEP for release. The scale bar in (b) is 5 μm. Reproduced with permission from [56]. Copyright 2018 by Springer Nature. (**C**) Dipolar interactions between Pt/SiO$_2$ Janus colloids and cargoes (PS). Cargoes will be released when the electric field is off. Reproduced with permission from [57]. Copyright 2018 by the Royal Society of Chemistry.

## 3.4. Thermophoresis

Slightly different is the case for heat-driven swimmers, which were first proposed by Würger theoretically [60] and then experimentally realized by Jiang [61]. Here, the absorption differences between the two Janus hemispheres trigger a thermophoretic mobility in the thin interfacial layer and subsequently particle motion. Despite several groups working on this type of swimmer [62,63], to this moment, only few active-active interaction studies have been reported, showing that at decreased inter-particle distances, the motion is dominated by inter-particle interactions, and clusters form. We are not aware of any published consideration of passive particles [64], but interactions that confirm the observations in other motion mechanisms can be expected [65].

## 3.5. Demixing: Casimir Forces

Despite the fact that demixing of a critical mixture is also caused by local heating, which pushes the system into its two-phase area, the microscopic interactions here are rather different, but similarities in observed behaviors are frequent, one example being wall interactions [66]. Here, the fuel is not consumed to cause swimmer motion, but a spinodal decomposition is happening, while interaction within the interfacial layer of the particles leads to propulsion without influencing the environment,

so that passive particles almost do not feel the vicinity of the swimmers via chemical gradients, which indicates that the interaction is purely hydrodynamic, except very close to the swimmer. This interaction was reported parenthetically by Buttinoni et al. [67] and later used to induce the formation of colloidal clusters, as well as their compression and surface melting (Figure 5B) [68]. In both cases, silica beads covered by a thin carbon layer in a critical mixture of 2,6-lutidine and water were observed under a light beam of 532 nm while the intensity was kept below 5 $\mu W/\mu m^2$ (Figure 5C). A slightly different system based on silica particles with iron oxide inclusions to absorb light and produce heat was used by Volpe's group (Figure 5A). An additional difference is the much higher light intensity, and combined, both parameters resulted in attractive interactions between the active and the passive particles, forming assemblies. The higher intensity of light caused a larger region of demixing, so that the passive colloids reacted with a phoretic response to the gradient [69].

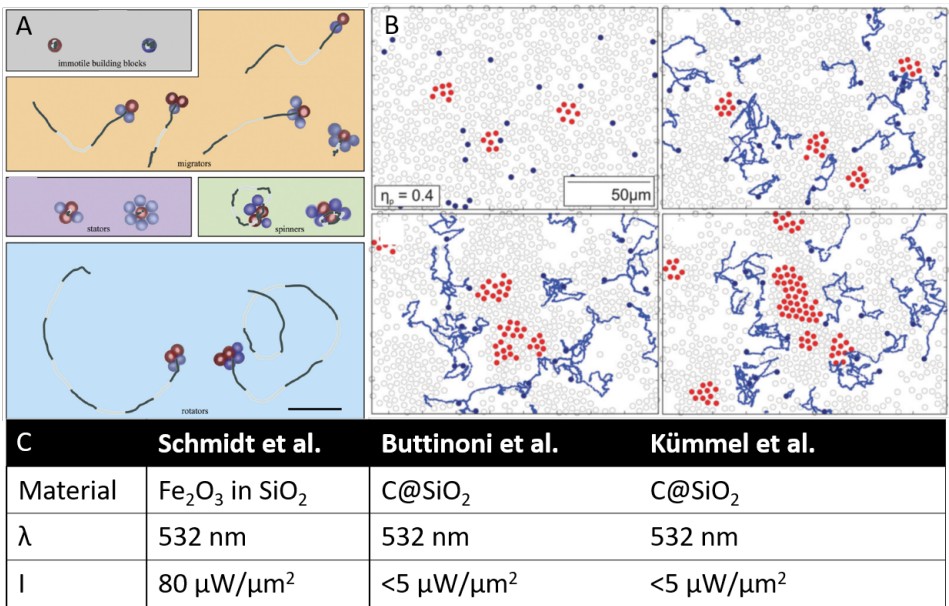

| C | Schmidt et al. | Buttinoni et al. | Kümmel et al. |
|---|---|---|---|
| Material | $Fe_2O_3$ in $SiO_2$ | $C@SiO_2$ | $C@SiO_2$ |
| λ | 532 nm | 532 nm | 532 nm |
| I | 80 $\mu W/\mu m^2$ | <5 $\mu W/\mu m^2$ | <5 $\mu W/\mu m^2$ |

**Figure 5.** (**A**) Different assemblies and movement behaviors based on immotile symmetric spherical particles, light-absorbing (brown), and non-absorbing (blue) particles. Dimers, trimers, and more complex axis-symmetric active molecules perform linear motion (migrators). Highly symmetric molecules do not show activity (stators). Symmetric chiral active molecules rotate almost without translating (spinners), and asymmetric chiral active molecules swim in circles (rotators). Lines represent particles' trajectories. Reproduced with permission from [69]. Copyright 2019 by AIP Publishing. (**B**) Experimental snapshots of the temporal evolution of a mixture of active and passive particles at 0, 600, 900, and 1200 s. Motion (blue trajectories) of active particles leads to the formation and compression of clusters of passive particles (marked in red). Reproduced with permission from [68]. Copyright 2015 by the Royal Society of Chemistry. (**C**) Materials, light wavelengths, and intensities for the above-mentioned works.

### 3.6. Chemical Gradient

Chemical electrolyte gradients released by a material itself allow the formation of microswimmers [70,71]. For instance, cation-exchange resin fragments (CIEX) create electrolyte gradients because of the different diffusivities of $H^+$ and $Cl^-$ ions. However, different from the Janus or bimetallic structures, which are inherently asymmetric, CIEX start to move when assembled with other inactive particles. The electrolyte gradient induces a long-range electro-osmotic flow, thus making passive colloidal particles approach the fragments. Once asymmetric colloid distribution has been formed, the complex starts moving (Figure 6A,B) [70]. However, with irregular CIEX, it is difficult to figure out the role of each component (CIEX and passive particles) precisely. Therefore,

regular spherical ion-exchange resins (IEX) were used to build a clear model [71]. Symmetry is broken when one passive particle in the range of electro-osmotic flow is advected toward the IEX. Moreover, passive particles not only contribute to the symmetry breaking, but also create additional electrophoretic flow. That can be proven by the velocity increase with the increase of a certain number particles (Figure 6C,D). Similar to Wang et al. [49], the authors also reported repulsive interactions when changing conditions [72].

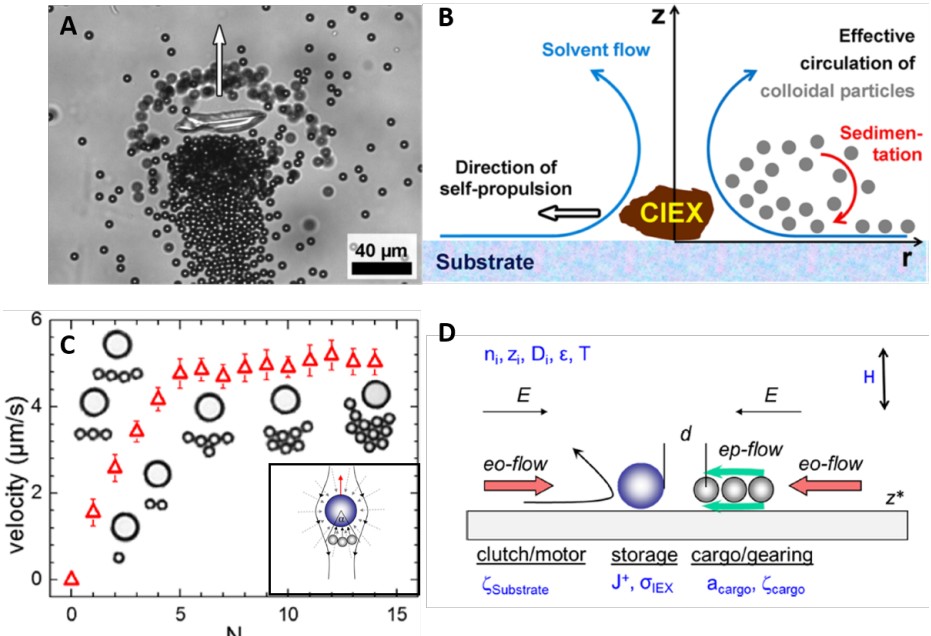

**Figure 6.** (**A**) Optical micrograph (top view) of a self-propelled complex consisting of an irregular cation-exchange resin fragments (CIEX) fragment and passive particles. (**B**) Scheme of the movement mechanism of the irregular complex. Reproduced with permission from [70]. Copyright 2013 by the American Chemical Society. (**C**) Velocity of complexes consisting of spherical IEX and passive particles versus the number of assembled cargo N. The inset is a qualitative sketch in top view with three cargoes assembled in the wake of an IEX bead. The dashed gray arrows indicate inward electro-osmotic flow (eo-flow), and the small solid arrows indicate the additional electrophoretic flow (ep-flow) arising from assembled passive particles (**D**) Scheme of the motion mechanism of assembling complex. Reproduced with permission from [71]. Copyright 2017 by the American Chemical Society.

### 3.7. Magnetic Forces

Even though some of the motion in this part is due to self-phoresis, a combination with magnetic effects can lead to other cooperative interactions. For example, magnetic interaction has been used for selective attraction of paramagnetic particles to diamagnetic particles (Figure 7A) [73] due to the magnetic interaction between magnetized micromotors and paramagnetic beads. However, the authors did not mention the range of that interaction. Nourhani et al. described in detail the use of contactless particle-particle interactions to push passive spherical magnetic tracers. Despite the fact that the propulsion of the active particles was normal diffusiophoresis, the interactions in this system seemed to be dominated by magnetic forces between the particles [74]. Steimel et al. discussed the range of interaction more in depth with two spinners immersed in passive particle media (Figure 7B) under a rotational magnetic field [75]. In that system, short-range magnetic dipole-dipole interaction was dominant when the distance between two spinners was smaller than 4D (D, diameter of particles). When the distance was larger than 4D, that interaction decayed fast and became too weak to attract. An ultra-long-range attractive interaction emerged in the presence of passive particle media. Elastic stresses induced by spinners were imposed on that dense passive monolayer; thus, an increase

of the interaction range between spinners up to 20D can be observed. Moreover, that ultra-long-range interaction can be modulated by the elasticity of the monolayer and the activity of the spinner, which are related to the area fraction of the passive monolayer and the rotational frequency of the magnetic field (Figure 7B,C), respectively. That makes it possible to control the range and magnitude of this kind of interaction in different systems. Adding silver islands to magnetically-driven helical swimmers, Ghosh et al. managed to microtrap plasmonically passive particles and use this system for transportation [76]. This elegant system allows very controlled manipulation of one or multiple passive microobjects, as well as transportation.

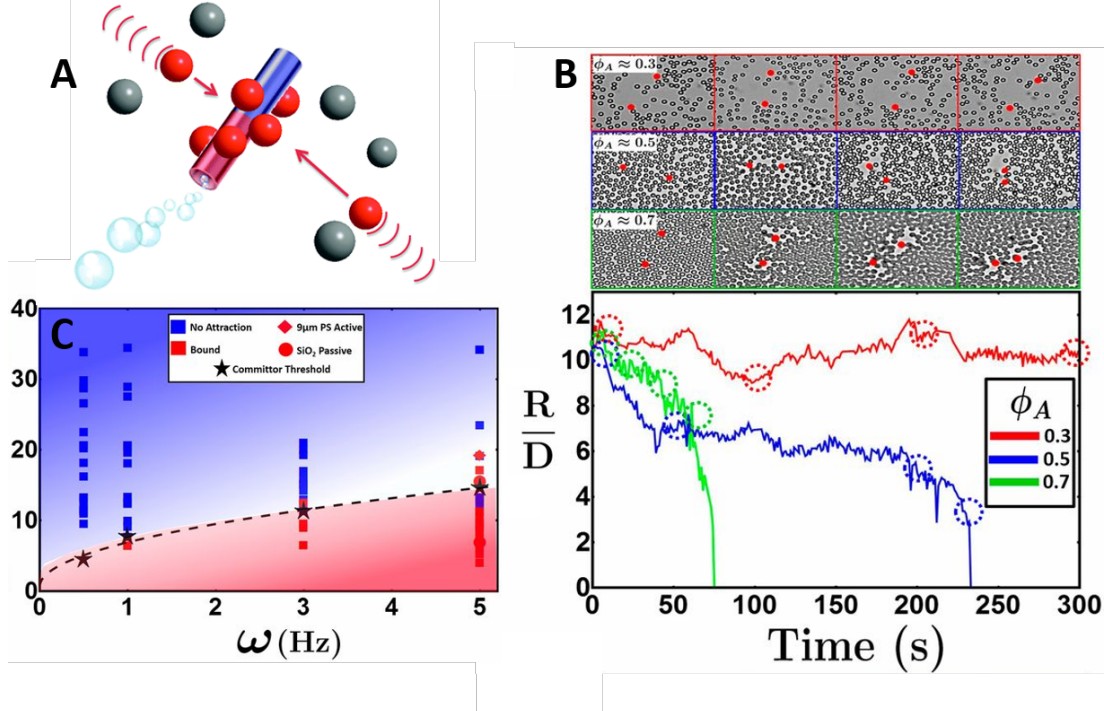

**Figure 7.** (**A**) Self-propelled micromotors containing a permanent magnetic moment selectively pick up paramagnetic particles (red) from diamagnetic particles (gray). Reproduced with permission from [73]. Copyright 2013 by the Royal Society of Chemistry. (**B**) Ultra-long-range interaction mediated by passive media. Experimental snapshots of two active spinners in passive mediums with different area fractions $\phi$A = 0.3 (red), 0.5 (blue), and 0.7 (green) and their distance trajectories with time. (**C**) Range of the interaction between spinners embedded in a dense colloidal monolayer of $\phi$A = 0.7 with different angular frequencies (blue symbols, no attraction; red symbols, bound). Reproduced with permission from [75]. Copyright 2016 by the National Academy of Sciences of the United States of America (NAS).

## 4. Applications

Within the literature, the phenomenon of phoretic interactions with passive particles has been reported and in some cases also studied more in depth. However, real applications or even steps towards them have rarely been mentioned, apart from cargo transport. Here, numerous proof of concept studies employing model colloids have been published [40,41,57], but concrete suggestions about where to use this property were only made by Wang et al., where light-driven micromotors could achieve the removal of different suspended microparticles. In very low peroxide concentrations, the titania-based microswimmers show, also due to their high speeds, significant phoretic attractions, which has been used to remove different materials from irradiated areas, among them microplastics extracted from daily care products, as well as from a contaminated aqueous medium [50]. From a fundamental point of view, this work deserves recognition because it proves the robustness of the phoretic interactions in terms of materials, shapes, and sizes. An even more precise control was achieved by Ghosh et al., using mobile optical traps grafted on helical microswimmers [76].

The absence of phoretic effects has been used to probe flow fields around biological microswimmers. Since no chemical fields are present, the interaction with tracer particles can be ascribed only to the fluid motion and is not influenced by local attraction or repulsion. Polin et al. used this effect on *Chlamydomonas* algae [35], and Drescher et al. looked at the flow fields around bacteria [77]. In the latter, the detection was rather difficult due to the small size of both the active objects and consequently also the tracers. Therefore, fluorescent bacteria themselves were used as flow indicators. However, trials to use particle image velocimetry (PIV) on artificial swimmers have been complicated by chemical fields causing the activity and leading to phoretic interactions. A team around Golestanian has demonstrated that by overlaying a large amount of data points of individual tracers interacting, PIV-type graphs can be obtained and fit well with theoretically-predicted flow fields [44].

A further interesting vision is to use interactions for the formation of new structures, as introduced by Fischer's group [48] and further developed by Palberg's group [69] and extensively reviewed and discussed by Niu et al. [71]. We envision that this strategy of active assembly and disassembly will have different applications to create novel, responsive materials in the future.

**Author Contributions:** Both authors planned, discussed and wrote the manuscript.

**Funding:** The authors thank the Volkswagen foundation for the Freigeist fellowship (Grant Number 91619), as well as the Kaercher foundation. L.W. would like to acknowledge the China Scholarship Council (CSC) for the financial support.

**Acknowledgments:** Interesting comments and discussions on the interactions and the manuscript with Ivo Buttinoni, Benno Liebchen, Hartmut Loewen, Falko Schmidt, Ian Madden, Erik Luijten, Mihail N. Popescu, William E. Uspal, Sandra Heckel, and Joost de Graaf are greatly acknowledged and for critical proofreading of the manuscript.

**Conflicts of Interest:** The authors declare no conflict of interest.

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
