# Peer review of "Review: Interactions of Active Colloids with Passive Tracers"

_condensedmatter, doi:10.3390/condmat4030078_

Round 1

Reviewer 1 Report

In this manuscript, Wang and Simmchen present a review of past theoretical and experimental work on interactions between active colloids and passive tracers. The study of active colloids (both from a theoretical and experimental point of view) has been a vibrant field of study in physics and chemistry over the past few years, with recent significant results. The attention of the community is now turning to the interaction of active colloids with their environment, and the question of interactions between active colloids and passive tracers is becoming more and more central. In this perspective, I acknowledge the effort made by the authors to summarize recent progress in this field. This review will surely find its audience and offer an interesting perspective. The experimental part of the paper (part 3) is informative, and I do not have major concerns about it.

However, in spite of this positive aspect, I have several concerns about the clarity, completeness and relevance of other sections of the paper. In particular, the part about theoretical considerations is extremely misleading. Some concepts are presented in a very elusive way, interactions between the active agents and their consequences are not presented clearly, and important references are lacking. Writing a complete and understandable review on the theoretical aspect would require a much longer discussion (as long as the experimental part). I believe that authors could drop this part altogether - and only mention theoretical approaches in their introduction, for instance. However, if they want to clarify this section and expand on it, here are my recommendations:
- A sketch of an active particle and its surrounding flow/chemical field could really help the reader
- Line 46 : the authors mention "a kind of phase transformation". This is extremely elusive. Surface reactions or physical transformations that yield phoretic transport are known in a very extensive and precise way, and need to be described with precise words and much more details.
- Generically, this section mixes results obtained with hydrodynamic interactions and without them. It makes the discussion very confused. The authors need to consider separately the study made with and without them, and then discuss the effect of hydrodynamic interactions on the active/passive colloids. For instance, Ref. [17] is irrelevant compared to the other references cited in the same paragraph. Some important references on the effect of hydrodynamic interactions on the collective behavior of active colloids are missing (see e.g. Matas-Navarro et al. Phys. Rev. E 90, 032304 (2014) or Zöttl and Stark Phys. Rev. Lett. 115, 118101 (2014), that should be cited). The last paragraph of the section mixes chemical interactions and hydrodynamic interactions and is not clear. The last sentence is quite vague and could be referenced.

Other major remarks:
- In Section 3.1, the authors should write that the enhanced diffusion of tracers in a bath of bacteria or active colloids was observed way before Refs. [25-28], see e.g. the seminal work by Wu and Libchaber [Phys. Rev. Lett. 84, 3017 (2000)].
- In the same Section, Ref. [26] should be discussed in another sentence. The results of Zöttl and Yeomans are about the mean speed of a swimming bacteria, and not about its diffusion. This is very different from a statistical perspective (measuring speed gives information about the mean position, whereas diffusion give information about the fluctuations around the mean position).
- In the same Section, Ref. [18] is presented in a very elusive way, and the sentence "The authors found that problems... to be expected" needs to be simplified. What is force-lattice coupling? The idea of [18] should be understandable without any technical and undefined concepts.
- There needs to be references in Section 3.2. Phoretic effects are very well-known both theoretically and experimentally, so the readers have to be oriented to the relevant bibliography. See e.g. the seminal review by Anderson [Ann. Rev. Fluid Mech. 21, 61 (1989)] or a more recent one by Illien, Golestanian and Sen [Chem. Soc. Rev. 46, 5508 (2017)].

Formatting and style of the manuscript:
- The quality of the figures is quite poor, the authors should enhanced it or enlarge the figures.
- There are a lot of typos and mistakes in the manuscript, which makes it troublesome to read. There are missing spaces on each line of the abstract. Calling references should be made before comas and full stops. Other typos (there are probably others I missed):
l. 41: self-phoretic
l. 51 and 52: use italic for mathematics
l. 71: "active" repeated
l. 152: "acive"
l. 156: "Well-defined"
l. 252 and later: repeated use of "don't" and "didn't". This is not correct in a manuscrit, use "do not" and "did not".
l. 322: extra comma after "We envision"
- The language is not always clear or appropriate, I would recommend the authors to ask a native English speaker to proofread their manuscript.

Author Response

Dear Reviewer, 

Thank you very much for getting these constructive reviews on our manuscript. 

Initially, we were puzzled by the bad marks used to evaluate the paper, despite the friendly words used to describe the impact of it. However, we found the comments very useful and agree that writing a complete and understandable review on the theoretical aspect would require a much longer discussion, which is not the scope of the paper.

However, we totally restructured the section and briefly discussed the categories dry and wet active matter, even though that still leaves certain parts elusive.  Writing a separate review on theory or dedicate as much space to theoretical approaches is not yet realistic, since most studies explicitly only take active active interactions into account. Since our focus was on comparing similar effects in different systems a basic description of all theoretical approaches would require not an article, but a whole book.

Anyway, the comments  helped to reorganize our ‘vision of the theoretical world’, which cost a lot of effort, but is priceless, thank you for the trigger.

All changes are marked in red in the manuscript. The references the reviewer suggested have been incorporated.

We appreciate the time and effort the referees have invested into commenting and improving the manuscript, especially also the many typos. 

Best regards, 

Juliane Simmchen

Reviewer 2 Report

The authors present a review article on interactions of active colloids with passive tracers. The article is well-compiled and provides an up-to-date description of the literature on this topic. I would recommend it for publication in your journal with a few minor corrections:

The title “Review Interactions of Active Colloids with Passive Tracers”  is missing a  hyphen “-“ or a preposition “on”  between the words Review and Interactions. Please rephrase it.

Lines 18-   A citation in the introduction part on active-passive interaction in biological medium would be helpful for the general reader.

Line 55-56 - Since  bacterias often do chemotaxis, the authors can include “self-generated chemical solutes” to distinguish artificial microswimmers.

Line 86-87 – The authors write “ difficult to obtain in real Janus particles and has raised some ongoing discussions” without any citation to a comment article on the cited paper. The authors should include citations for this claim or simply highlight the experimental challenges.

Line 119-   Please mention the year instead of stating “In early times”

Line 164 -  Including  an inline citation alongwith “Palacci’s hematite composites”, would be helpful for the  general reader, since the citation comes much later in line 170.

Section 3.7- Among magnetic interaction section, the authors should also cite the article on   “contactless particle-particle interactions”, that has been used to push passive tracers with active colloids in the paper.

Engineering Contactless Particle–Particle Interactions in Active Microswimmers

Amir Nourhani Daniel Brown Nicholas Pletzer  John G. Gibbs

https://doi.org/10.1002/adma.201703910

Line 292-293:  “ Adding gold particles to magnetically driven helical swimmers, Ghosh et al. managed to optically microtrap passive particles and use this system for transportation.”

According to the paper,  “gold” should be “silver islands” and it is a  “optically controlled plasmonic trap”. Same goes for line 308-  “mobile optical trap” should be “mobile plasmonic trap”

Line 305:  In my opinion, “contaminated river” sounds bit a far-stretched, “contaminated aqueous medium” would be more appropriate.

Typohraphical errors:

Typographical errors- There are a few type-setting errors in the abstract. Please revise the abstract and include space in words like “interestsummarizing” , “Forthat”, “theoreticalworks” etc.

Line 26 - “authors” should be “author’s”

Line 220,224 -  “dielectricphoretic” should be “dielectrophoretic”

Line 278 -  “Cooperate” should be “cooperative”

Author Response

Dear Reviewer, 

Thank you very much for getting these constructive reviews on our manuscript

All changes are marked in red in the manuscript, all references that were  suggested have been incorporated:

- The title was rephrased.

- Further citations were added into the introduction. Besides, due to some comments of the other reviewer we restructured also the theoretical part. 

- the addition “self-generated chemical solutes” was added to the manuscript to facilitate reading. 

- we added the citation to the comment, it will be published in its final version very soon. 

- we replaced 'in early times' by the year.

- We referenced  “Palacci’s hematite composites” earlier in the text but kept also the latter citation 

- Thank you very much for pointing out that we were missing the reference on the overlaying magnetic influences. We find it very curious, that magnetic forces are dominating the attractive phoretic interactions that should be prevailing in the Pt driven system. 

- we corrected the terms referring to Ambarish Ghosh paper. 

- we replaced  “contaminated river” by  “contaminated aqueous medium” and thank the reviewer for paying attention, we do not want to raise expectations we can not fulfill.

In genereal, we appreciate the time and effort the referees have invested into commenting and improving the manuscript, especially also the many typos, which have been corrected. 

Best regards, 

Juliane Simmchen

Round 2

Reviewer 1 Report

The authors have significantly improved the theoretical section of the manuscript. It is now very well organized, and contains relevant references. This considerably improves the quality of the review.

I also acknowledge the efforts to correct typos and change some phrasings.

I may now recommend this review for publication.